# Kaempferol 3-Rhamnoside on Glutamate Release from Rat Cerebrocortical Nerve Terminals Involves P/Q-Type Ca^2+^ Channel and Ca^2+^/Calmodulin-Dependent Protein Kinase II-Dependent Pathway Suppression

**DOI:** 10.3390/molecules27041342

**Published:** 2022-02-16

**Authors:** Tzu-Kang Lin, Chi-Feng Hung, Jing-Ru Weng, Ting-Yang Hsieh, Su-Jane Wang

**Affiliations:** 1Department of Neurosurgery, Fu Jen Catholic University Hospital, Fu Jen Catholic University, New Taipei City 24205, Taiwan; tklin100@gmail.com; 2School of Medicine, Fu Jen Catholic University, New Taipei City 24205, Taiwan; skin@mail.fju.edu.tw; 3Department of Marine Biotechnology and Resources, National Sun Yat-Sen University, Kaohsiung 80424, Taiwan; jrweng@mail.nsysu.edu.tw; 4P.H.D. Program in Nutrition & Food Science, Fu Jen Catholic University, New Taipei City 24205, Taiwan; david711young@gmail.com; 5Research Center for Chinese Herbal Medicine, College of Human Ecology, Chang Gung University of Science and Technology, Taoyuan 33303, Taiwan

**Keywords:** kaempferol 3-rhamnoside, glutamate release, P/Q-type Ca^2+^ channels, Ca^2+^/calmodulin-dependent protein kinase II, cerebrocortical nerve terminals

## Abstract

Excess synaptic glutamate release has pathological consequences, and the inhibition of glutamate release is crucial for neuroprotection. Kaempferol 3-rhamnoside (KR) is a flavonoid isolated from *Schima superba* with neuroprotective properties, and its effecton the release of glutamate from rat cerebrocortical nerve terminals was investigated. KR produced a concentration-dependent inhibition of 4-aminopyridine (4-AP)-evoked glutamate release with half-maximal inhibitory concentration value of 17 µM. The inhibition of glutamate release by KR was completely abolished by the omission of external Ca^2+^ or the depletion of glutamate in synaptic vesicles, and it was unaffected by blocking carrier-mediated release. In addition, KR reduced the 4-AP-evoked increase in Ca^2+^ concentration, while it did not affect 4-AP-evoked membrane potential depolarization. The application of selective antagonists of voltage-dependent Ca^2+^ channels revealed that the KR-mediated inhibition of glutamate release involved the suppression of P/Q-type Ca^2+^ channel activity. Furthermore, the inhibition of release was abolished by the calmodulin antagonist, W7, and Ca^2+^/calmodulin-dependent protein kinase II (CaMKII) inhibitor, KN62, but not by the protein kinase A (PKA) inhibitor, H89, or the protein kinase C (PKC) inhibitor, GF109203X. We also found that KR reduced the 4-AP-induced increase in phosphorylation of CaMKII and its substrate synapsin I. Thus, the effect of KR on evoked glutamate release is likely linked to a decrease in P/Q-type Ca^2+^ channel activity, as well as to the consequent reduction in the CaMKII/synapsin I pathway.

## 1. Introduction

Glutamate is the primary excitatory neurotransmitter in the central nervous system (CNS) and is responsible for normal brain function [1]. Therefore, maintaining brain glutamate homeostasis is crucial to ensure proper glutamatergic neurotransmission and neuronal viability [2,3]. Indeed, excess glutamate at synapses leads to neuronal damage through the overactivation of glutamate receptors and high Ca^2+^ influx, contributing to the etiopathogenesis of numerous neurological disorders, including ischaemia, epilepsy, and neurodegenerative diseases [4,5]. In this context, the compounds inhibiting synaptic glutamate release could be therapeutic for disease treatment because they counteract hyperglutamatergic effects, participating in the restoration of glutamate transmission in the CNS [6,7,8]. Natural compounds derived from medicinal plants are an important sourceof potential molecules for therapeutic application. For example, it has been reported that baicalein (a flavonol of *Scutellariabaicalensis*), 11-keto-*β*-boswellic acid (a triterpenoid of *Boswellia serrata*), and asiatic acid (a triterpene of *Centella asiatica*) can reduce synaptic glutamate release and thereby protect neurons [9,10,11,12].

Kaempferol 3-rhamnoside (KR, Figure 1A) is a flavonol glycoside extracted from the bark of *Schima superba*, which is a broad-leaved evergreen tree of the Camellia family, and widely used in traditional Chinese medicine for heat clearing, detoxification, and furuncle treatment [13]. The antitumor, anti-inflammatory, antioxidant, antibacterial, anticandidal, antiobesity, and hepatoprotective properties of KR have been documented in several bioassays [14,15,16,17,18,19]. However, the biological targets and effects of KR on the CNS are largely unknown. To our knowledge, to dateonly one study has reported that KR ameliorates synaptic plasticity and memory functionin a mouse model of scopolamine-induced amnesia [20]. This result suggests that KR could be a promising neuroprotective agent. Excessive glutamate release has pathological consequences, and the inhibition of glutamate release is crucial for neuroprotection [4,6,7]. Therefore, the present study used isolated nerve terminals of the rat cerebral cortex to investigate the effect of KR on glutamate release and the possible mechanism involved. Isolated nerve terminals carry the structural features and properties of the in vivo neuronal terminals from which they originate. They are widely recognized as an in vitro model for studying presynaptic regulation of neurotransmitter release [21,22]. In the present work, the efficacy of KR on endogenous glutamate release, cytosolic Ca^2+^ levels, membrane potential, and the phosphorylation of protein kinase was investigated.

## 2. Results

### 2.1. KR Inhibitsthe Release of Glutamate Evoked by 4-AP atRat Cerebrocortical Nerve Terminals

To analyze whetherKR affected synaptic glutamate release, isolated nerve terminals (synaptosomes) were stimulated with the K^+^ channel blocker, 4-AP. 4-AP destabilizes the membrane potential and is thought to cause repetitive spontaneous Na^+^ channel-dependent depolarization that closely approximates in vivo depolarization of the synaptic terminal, which leads to the activation of voltage-dependent Ca^2+^ channels (VDCCs) and neurotransmitter release [23]. As presented in Figure 1B, stimulation of synaptosomes with 4-AP (1 mM) for 5 min evoked glutamate release of 7.5 ± 0.1 nmol/mg/5 min in the presence of 1.2 mM CaCl_2_. The synaptosomes were incubated with KR (30 µM) for 10 min before the addition of 4-AP decreased 4-AP-evoked glutamate release to 3.0 ± 0.2 nmol/mg/5 min (*n* = 5, *p* < 0.001), without affecting the basal release of glutamate. The IC_50_ for this inhibition was 17 µM, and the maximum effect was obtained at 30 µM (Figure 1C).

In nerve terminals, 4-AP induces glutamate release through Ca^2+^-dependent vesicular release and reversal of glutamate uptake transporters (Ca^2+^-independent release) [24]. Because KR decreased glutamate release by synaptosomes evoked by 4-AP, we analyzed glutamate release following the addition of 300 µM EGTA and the chelation of extracellular Ca^2+^ before depolarization with 4-AP. This cytosolic release of glutamate amounted to less than 2 nmol/mg/5 min (*n* = 5, *p* < 0.001). Notably, glutamate release was not affected by KR (30 µM) pretreatment (*p* = 0.55; Figure 1D). Furthermore, we used either DL-threo-β-benzyloxyaspartate (DL-TBOA), a nonselective inhibitor of all excitatory amino acid transporter subtypes, or bafilomycin A1, a vacuolar H^+^ ATPase inhibitor that causes the depletion of glutamate in synaptic vesicles, to test the effect of KR. The addition of DL-TBOA (10 µM) before the addition of 4-AP increased 4-AP-evoked glutamate release (*n* = 5, *p* < 0.001). When DL-TBOA was present, KR (30 µM) still effectively decreased 4-AP-evoked glutamate release (*p* < 0.001; Figure 1D). In contrast, bafilomycin A1 (0.1 µM) decreased 4-AP-evoked glutamate release (*n* = 5, *p* < 0.001). When bafilomycin A1 was present, the inhibitory effect of KR on 4-AP-evoked glutamate release was strongly attenuated. The glutamate release measured in the presence of both bafilomycin A1 and KR resembled that obtained in the presence of bafilomycin A1 alone (*p* = 0.79; Figure 1D). These results suggest that the inhibition of glutamate release by KR affects the Ca^2+^-dependent exocytotic component of 4-AP-evoked glutamate release.

### 2.2. KR Reduces the 4-AP-Induced Increase in [Ca^2+^]_C_

To confirm this hypothesis, the action of KR on cytosolic Ca^2+^ levels ([Ca^2+^]_C_) in synaptosomes was tested. As presented in Figure 2A, the addition of 4-AP (1 mM) increased [Ca^2+^]_C_ from 171.4 ± 1.9 nM to a plateau of 236.9 ± 3.2 nM. Preincubation of synaptosomes with KR (30 µM) did not significantly affect basal Ca^2+^ levels (168.7 ± 2.6 nM; *p* = 0.42), but reduced the 4-AP-evoked [Ca^2+^]_C_ increase by 21% (186.3 ± 2.8 nM; *n* = 5, *p* < 0.001). In addition, KR significantly reduced 15 mM KCl-evoked glutamate release, which involves only VDCC activation [23] (*n* = 5, *p* < 0.001; Figure 2B).

### 2.3. KR Does Not Affect Synaptosomal Membrane Potential

To understand whether the inhibitory effect of KR on glutamate release was due to a decrease in synaptic excitability, the effect of KR on the synaptosomal plasma membrane potential was examined using the membrane potential-sensitive dye DiSC_3_(5). As presented in Figure 3, the addition of 4-AP (1 mM) caused an increase in DiSC_3_(5) fluorescence in synaptosomes from 0.2 ± 0.2 to 26.9 ± 0.6 fluorescence units/5 min.The preincubation of synaptosomes with KR (30 µM) did not alter the basal DiSC_3_(5) fluorescence (*p* = 0.37), and produced no substantial change in the 4-AP-mediated increase in DiSC_3_(5) fluorescence (*n* = 5, *p* = 0.21), excluding the involvement of reduced membrane potential.

### 2.4. KR-Mediated Inhibition of 4-AP-Evoked Glutamate Release Is Abolished by P/Q-Type VDCCs Blockade

Because glutamate release is supported by N- and P/Q-type VDCCs at rat cerebrocortical nerve terminals [25,26], the involvement of specific VDCCs in the KR inhibition of 4-AP-evoked glutamate release was investigated using Ca^2+^ channel blockers. As presented in Figure 4, control glutamate release evoked by 4-AP was reduced by KR from 7.7 ± 0.1 nmol/mg/5 min to 2.8 ± 0.1 nmol/mg/5 min (*p* < 0.001). Likewise, ω-conotoxin GVIA (ω-CgTX GVIA), a blocker of N-type Ca^2+^ channels, reduced 4-AP-evoked glutamate release (*n* = 5, *p* < 0.001); however, KR could still effectively reduce release in the presence of ω-CgTX-GVIA (*p* < 0.05). Although ω-agatoxin IVA (ω-Aga IVA), a blocker of the P/Q-type Ca^2+^ channels, produced a stronger reduction in 4-AP-evoked glutamate release (*n* = 5, *p* < 0.001), further inhibition by KR could not be observed in this case (*p* = 0.98). In addition, we examined whether the inhibition of 4-AP-evoked glutamate release by KR was due to a decrease in intracellular Ca^2+^ release [27]. As presented in Figure 4, dantrolene (10 µM), an inhibitor of intracellular Ca^2+^ release from the endoplasmic reticulum, and 7-chloro-5-(2-chlorophenyl)-1,5-dihydro-4,1-benzothiazepin-2(3H)-one (CGP37157) (10 µM), an inhibitor of mitochondrial Na^+^/Ca^2+^ exchange, reduced 4-AP-evoked glutamate release (*n* = 5, *p* < 0.001). However, KR (30 µM) could still effectively reduce glutamate release in the presence of dantrolene or CGP37157 (*p* < 0.001). These data point to the possibility that the action of KR is characterized through the suppression of P/Q-type Ca^2+^ channel activity.

### 2.5. The Suppressed Ca^2+^/Calmodulin-Dependent Protein Kinase II Pathway Is Linked to KR-Mediated Inhibition of 4-AP-Evoked Glutamate Release

To identify whether the intraterminal enzymatic pathways participate in the KR inhibition of glutamate release, we tested several enzyme inhibitors: N-[2-(p-bromocinnamylamino)ethyl]-5-isoquinolinesulfonamide (H89), a protein kinase A (PKA) inhibitor; bisindolylmaleimide I (GF109203X), a PKC inhibitor; 2-(2- amino-3-methoxyphenyl)-4H-1-benzopyran-4-one (PD98059), a mitogen-activated protein kinase (MAPK) inhibitor; and 1-[N, O-bis(5-Isoquinolinesulfonyl)-N-methylL-tyrosyl]-4-phenylpiperazine (KN62), an inhibitor of Ca^2+^/calmodulin-dependent protein kinase II (CaMKII). As presented in Figure 5, control glutamate release evoked by 4-AP was significantly reduced by KR (30 µM) (*n* = 5, *p* < 0.001). Control glutamate release evoked by 4-AP was also significantly reduced by H89 (100 µM), GF109203X (10 µM), PD98059 (50 µM), and KN62 (10 µM) (*n* = 5, *p* < 0.001). However, in the presence of H89, GF109203X, or PD98059, 4-AP-evoked glutamate release was further reduced by KR (*p* < 0.001). In contrast, KR inhibition of 4-AP-evoked glutamate release was abolished in the presence of KN62. Glutamate release measured in the presence of both KN62 and KR was similar tothat obtained in the presence of KN62 alone (*p* = 0.99; Figure 5). Furthermore, N-(6-aminohexyl)-5-chloro-1- naphthalene-sulphonamide (W7, 5 µM), a calmodulin antagonist, also reduced 4-AP-evoked glutamate release (*p* < 0.001). However, further inhibition by KR was not observed in the presence of W7 (*n* = 5, *p* = 0.83). These results indicate that the inhibitory effect of KR on 4-AP-evoked glutamate release is linked to the suppression of CaMKII, but not PKA, PKC or MAPK activity.

### 2.6. KR Reduces the Phosphorylation Levels of CaMKII and Synapsin I

To confirm the effect of KR on CaMKII activity, we examined the expression and phosphorylation levels of CaMKII and its substrate synapsin I at cerebrocortical nerve terminals using Western blotting. As presented in Figure 6, the addition of 4-AP did not influence the expression levels of CaMKII and synapsin I (*p* = 0.5), but it increased the phosphorylation levels of CaMKII and synapsin I (Ser 603) (*n* = 5, *p* < 0.001). When synaptosomes were pretreated with KR (30 µM) for 10 min before the addition of 4-AP, 4-AP-induced phosphorylation of CaMKII and synapsinI was significantly decreased compared with that in the 4-AP group (*p* < 0.001).

## 3. Discussion

Excessive glutamate in the synaptic cleft is one of the main causes of neuronal death in neurological disorders [4]. Limiting excess synaptic glutamate release is a potential target for neuroprotectants [6,7]. Many natural products have been studied for their neuroprotective effects, and they are a valuable source for developing new neuroprotectants [28,29]. Among them, flavonoids are important and widely distributed natural compounds with various biological effects that have been found to act by presynaptically inhibiting glutamate release [10]. KR, also known as kaempferin or afzelin, is aflavonol glycoside isolated from the leaves of *Schima superba*, and has been reported to have neuroprotective activity [20]. However, to our knowledge, to date no studies investing the action of KR on glutamate release have been reported. Thus, the present study investigated the effect of KR on glutamate release at rat cerebrocortical nerve terminals, and assessed the possible mechanism.

In the present study, KR inhibited 4-AP-evoked glutamate release from synaptosomes in a concentration-dependent manner with an IC_50_ value of 17 µM. Regarding 4-AP-induced glutamate release from neurons, two mechanisms (exocytosis and carrier-mediated outward transport) are involved. Exocytotic glutamate is released from stored vesicles (Ca^2+^-dependent fraction), whereas glutamate transporters transport glutamate from the axoplasmic site via a reduced Na^+^ gradient (Ca^2+^-dependent fraction) [23,24]. We found that KR reduced 4-AP-evoked glutamate release in the presence of extracellular Ca^2+^; however, it had no effect in the absence of extracellular Ca^2+^. Furthermore, the depletion of glutamate in synaptic vesicles by bafilomycin A1 largely prevented the inhibition of glutamate release by KR; however, the inhibition of release by KR was insensitive to blockade of the glutamate transporter by DL-TBOA. These results indicate that KR decreases 4-AP-evoked glutamate release at cerebrocortical nerve terminals by inhibiting Ca^2+^-dependent exocytotic release of vesicular glutamate rather than inhibiting reversal of glutamate uptake transporters.

KR-mediated inhibition of Ca^2+^-dependent glutamate release occurs by reducing synaptosomal plasma membrane potential, which may indirectly lead to attenuation of VDCC activity [23,30]. However, we found that at cerebrocortical nerve terminals, KR reduced the 4-AP-evoked increase in [Ca^2+^]_C_, while it did not affect 4-AP-mediated depolarization, indicating that the observed inhibition by KR is not due to a change in membrane potential. Additionally, apart from 4-AP-evoked glutamate release, KR also reduced KCl-evoked glutamate release. Because 4-AP-evoked glutamate release involves the activation of Na^+^ and Ca^2+^ channels, 15 mM external KCl-evoked glutamate release involves only Ca^2+^ channels [23,31], which indicates that the inhibition of 4-AP-evoked glutamate release by KR appears to act through a direct suppression of VDCC activity. This suggestion was supported by the observation that the KR-mediated inhibition of release was completely abolished after exposure to the P/Q-type Ca^2+^channel blocker ω-AgTX-IVA. However, the KR-mediated inhibition of glutamate release was unaltered by pretreatment with the N-type Ca^2+^ channel blocker ω-CgTX-GVIA and the intracellular Ca^2+^ release inhibitors dantrolene and CGP37157. Based on our data, we demonstrate that KR inhibits P/Q-type Ca^2+^ channel-dependent glutamate release. Although how KR influences P/Q-type Ca^2+^ channels remain unclear, two possibilities might be considered: direct inhibition of KR on Ca^2+^ channel, and dependence on the interaction of KR with presynaptic proteins [32].

The present study also supports a role for decreased CaMKII activity in the KR inhibition of glutamate release. This stems from the following observations: (1) the inhibitory effect of KR on 4-AP-evoked glutamate release was prevented by calmodulin and CaMKII inhibition but not by PKA and PKC inhibition; and (2) KR significantly decreased 4-AP-induced phosphorylation of CaMKII and synapsin I, a major presynaptic substrate for CaMKII. CaMKII is known to modulate the exocytotic pathway at nerve terminals, where it is activated by Ca^2+^/calmodulin, and increases neurotransmitter release through phosphorylation of several proteins of the exocytotic machinery, including synapsin I [33,34]. Synapsin I phosphorylation results in increased synaptic vesicle availability for release [35]. Thus, we infer that at cerebrocortical nerve terminals, KR decreases Ca^2+^ influx through P/Q-type Ca^2+^ channels, which causes a subsequent decrease in CaMKII activation and synapsin I phosphorylation, and a consequent reduction in glutamate release (Figure 7).

## 4. Materials and Methods

### 4.1. Isolation of KR

The bark of *Schima superba* (2.0 kg) wasground, extractedwith methanol at room temperature, and concentrated under reduced pressure to form a brown residue (110 g).This residue was partitioned using EtOAc/H_2_O, and the EtOAc layer (99 g) was fractionated by silica gel column chromatography using *n*-hexane-EtOAc, 9:1; *n*-hexane-EtOAc, 4:1; *n*-hexane-EtOAc, 1:1; and *n*-hexane-EtOAc-MeOH, 1:1:1, to yield fourfractions (A–D).Fraction B (102 mg), CH_2_Cl_2_-MeOH (4:1) to provide fractions B_1_–B_6_, fraction B_3_ (55 mg),CHCl_3_-MeOH (4:1) to yield KR(10 mg).The identity and purity of KR were verified by proton nuclear magnetic resonance (NMR) spectroscopyand 2-D NMR spectrometry [36].

### 4.2. Chemicals

DL-TBOA, bafilomycin A1, dantrolene, CGP37157, KN62, H89, and GF109203X were purchased from Tocris (Bristol, UK). 3,3,3-Dipropylthiadicarbocyanine iodide [DiSC_3_(5)] and fura-2-acetoxymethyl ester (Fura-2-AM) were purchased from Thermo (Waltham, MA, USA).ω-CgTX GVIA and ω-Aga IVA were purchased from Alomone lab (Jerusalem, Israel). 4-AP, dimethylsulfoxide (DMSO), and all other reagents were purchased from Sigma-Aldrich (St. Louis, MO, USA).

### 4.3. Animals

All the experiments were carried out using male SpragueDawley rats weighing 150–200 g (*n* = 30) purchased from BioLASCO (Taipei, Taiwan). The experimental protocol was approved by the Fu Jen Institutional Animal Care and Utilization Committee with code A10925. The animals were treated in accordance with the Guide for the Care and Use of Laboratory Animals. The minimal number of animals to obtain consistent data were employed. The rats were sacrificed via cervical dislocation to prepare the synaptosomes for endogenous glutamate release assay (*n* = 15), cytosolic Ca^2+^ assay (*n* = 5), membrane potential assay (*n* = 5), and Western blotting (*n* = 5).

### 4.4. Preparation of Isolated Nerve Terminals

The preparation of cerebrocortical nerve terminals from the rats was performed as previously described [9,37]. Briefly, the rats were sacrificed, and the cerebral cortex of each rat wasremoved immediately. The brain tissue was homogenized in ice-cold sucrose solution and centrifuged at 3000× *g* for 10 min at 4 °C. The supernatant was then centrifuged at 14,500× *g* for 12 min at 4 °C, and the pellet resuspended on Percoll discontinuous gradients (3%, 10%, and 23% Percoll). After centrifugation at 32,500× *g* for 7 min at 4 °C, the synaptosomal fraction placed between the 10% and the 23% Percoll was collected and diluted in HEPES buffer mediumconsisting of 140 mM NaCl, 5 mM KCl, 5 mM NaHCO_3_, 1 mM MgCl_2_-6H_2_O, 1.2 mM Na_2_HPO_4_, 10 mM glucose, and 10 mM HEPES at pH 7.4. After centrifugation at 27,000× *g* for 10 min, the protein content was determined using the Bradford assay. Synaptosomes were centrifuged in a final wash to obtain synaptosomal pellets with 0.5 mg protein.

### 4.5. Endogenous Glutamate Release Assay

The synaptosomal pellet was resuspended in the HEPES buffer medium, and glutamate release was assayed by online fluorimetry [37]. Briefly, synaptosomal pellets were resuspended in a HEPES buffer medium containing CaCl_2_ (1.2 mM), glutamate dehydrogenase (GDH, 50 units/mL), and NADP^+^ (2 mM), in a Perkin-Elmer LS-50B spectrofluorimeter (Beaconsfield, UK). Glutamate release was evoked with 4-AP (1 mM) and monitored by measuring the increase in fluorescence (excitation and emission wavelengths of 340 and 460 nm, respectively) resulting from NADPH being produced by the oxidative deamination of released glutamate by GDH. Released glutamate was calibrated by a standard of exogenous glutamate (5 nmol) and expressed as nanomoles glutamate per milligram synaptosomal protein (nmol/mg). Values quoted in the text and expressed in the bar graphs represent levels of glutamate cumulatively released after 5 min of depolarization, and are expressed as nmol/mg/5 min.

### 4.6. Cytosolic Ca^2+^Assay

The synaptosomal pellet was incubated in the HEPES buffer medium containing Fura 2-AM (5 μM) and CaCl_2_ (0.1 mM) for 30 min at 37 °C. Synaptosomes were centrifuged for 1 min at 3000× *g*, and pellets were resuspended in HEPES buffer medium containing CaCl_2_ (1.2 mM). Fura-2-Ca fluorescence was determined in a Perkin-Elmer LS-55spectrofluorimeter with excitation wavelengths of 340 and 380 nm (emission wavelength 505 nm) at 5 s intervals for 5 min. Calibration procedures were performed as described by previously [38], using 0.1% sodium dodecyl sulphate to obtain the maximal fluorescence with Fura-2 saturation with Ca^2+^, followed by 10 mM EGTA (Tris buffered) to obtain minimum fluorescence in the absence of any Fura-2/Ca^2+^ complex. Cytosolic free Ca^2+^ concentration ([Ca^2+^]_c_, nM) was calculated using equations described by previous studies [39].

### 4.7. Membrane Potential Assay

The synaptosomal membrane potential was assayed with the positively charged membrane potential-sensitive carbocyanine dye DiSC_3_(5). The dye becomes incorporated into the synaptosomal plasma membrane lipid bilayer. Upon depolarization with 4-AP, the release of the dye from the membrane bilayer is indicated by an increase in fluorescence [40]. The synaptosomal pellet was incubated in the HEPES buffer medium containing 1.2 mM CaCl_2_, 5 mM DiSC_3_(5) and 1 mM 4-AP for 10 min at 37 °C. Fluorescence was determined in a Perkin-Elmer LS-55spectrofluorimeter with excitation and emission wavelengths set at 646 and 674 nm, respectively.

### 4.8. Western Blot Assay

Synaptosomes were homogenized in a lysis buffer and centrifuged at 12,000× *g* for 20 min. The protein content was determined using the Bradford assay (Bio-Rad laboratories, Hercules, CA, USA). Equal amounts of protein samples (20 μg) were separated by 10% SDS-polyacrylamide gel electrophoresis and transferred to polyvinylidene difluoride membranes (Bio-Rad Laboratories). Transferred membrane was blocked with 5% nonfat dry milk in Tris-buffered saline/Tween 20 solution. The blots were incubated with CaMKII (1:10,000, Cell Signaling, Beverly, MA, USA), phospho-CaMKII (1:2000, Cell Signaling, Beverly, MA, USA), synapsin I (1:100,000, Cell Signaling, Beverly, MA, USA), and phospho-synapsin I (Ser 603) (1:2000, GeneTex, Irvine, CA, USA) at 4 °C overnight. β-actin (1:8000, Cell signaling, Beverly, MA, USA) was performed as an internal control. After washing with Tris-buffered saline/Tween 20 solution, horseradish peroxidase-conjugated secondary antibodies (1:16,000, Gentex, GTX213110-01, GTX213111-01, Zeeland, MI, USA) were applied, and the blots were developed using the enhanced chemiluminescence detection system (Amersham Biosciences Corp., Amersham, Buckinghamshire, UK). Quantification was obtained by scanning densitometry of five independent experiments, and analyzed with Image J software (Synoptics, Cambridge, UK).

### 4.9. Data and Statistical Analysis

Results are presented as mean ± S.E.M. Statistical analysis was performed using Prism version 5.0 (Graph Pad Software Inc., San Diego, CA, USA). When testing the significance of the effect of KR versus control, Student’s *t*-test was used. When comparing the effect of KR in different experimental conditions, one-way analysis of variance (ANOVA) was used, followed by Dunnett’s multiple-comparison test. *p* values of 0.05 or less were considered to represent significant differences.

## 5. Conclusions

This is the first study to demonstrate that KR has an inhibitory effect on glutamate release in rat cerebrocortical nerve terminals. This effect might be exerted mainly through the suppression of P/Q-type Ca^2+^ channels and CaMKII/synapsin I pathways. Our study stimulates its application in preclinical studies directed to the development of a new drug for brain diseases related to glutamate excitotoxicity.

## Figures and Tables

**Figure 1 molecules-27-01342-f001:**
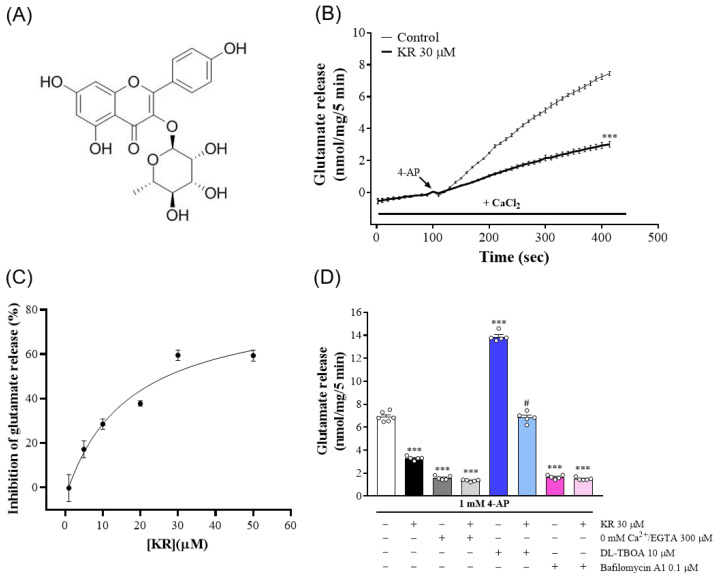
Effect of KR on 4-AP-evoked glutamate release at rat cerebrocortical nerve terminals. (**A**) Chemical structure of KR. (**B**) Time course of glutamate release evoked by 4-AP (1 mM) in the absence of added drug (control), or in the presence of KR added 10 min before the addition of 4-AP. (**C**) Concentration-response curve for the effect of KR on 4-AP-evoked glutamate release. (**D**) Effects of 0 mM Ca^2+^/EGTA, glutamate transporter inhibitor DL-TBOA and vacuolar H^+^ ATPase inhibitor bafilomycin A1 on 4-AP-evoked glutamate release in the absence or in the presence of KR. Each point or bar represents the mean ± S.E.M. of results obtained in 5 experiments. *** *p* < 0.001 versus control; # *p* < 0.001 versus DL-TBOA alone.

**Figure 2 molecules-27-01342-f002:**
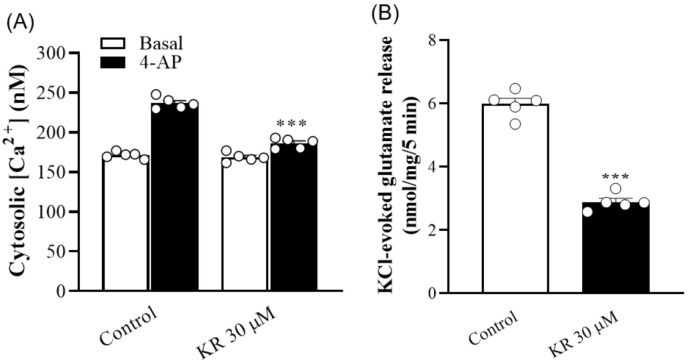
(**A**) Effect of KR on cytosolic Ca^2+^ concentration [Ca^2+^]_C_. (**B**) Effect of KR on KCl (15 mM)-evoked glutamate release. Each bar represents the mean ± S.E.M. of results obtained in 5 experiments. *** *p* < 0.001 versus control.

**Figure 3 molecules-27-01342-f003:**
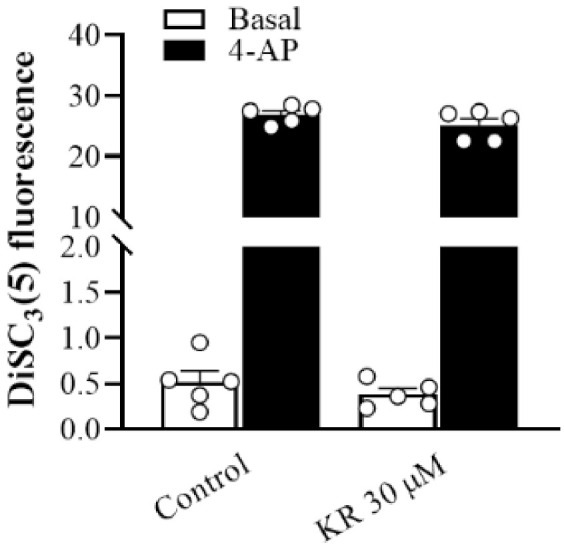
Effect ofKR on synaptosomal membrane potential. Synaptosomal membrane potential was monitored with DiSC_3_(5) on depolarization with 1 mM 4-AP, in the absence (control) or presence of 30 µM KR added 10 min before depolarization. Each bar represents the mean ± S.E.M. of results obtained in 5 experiments.

**Figure 4 molecules-27-01342-f004:**
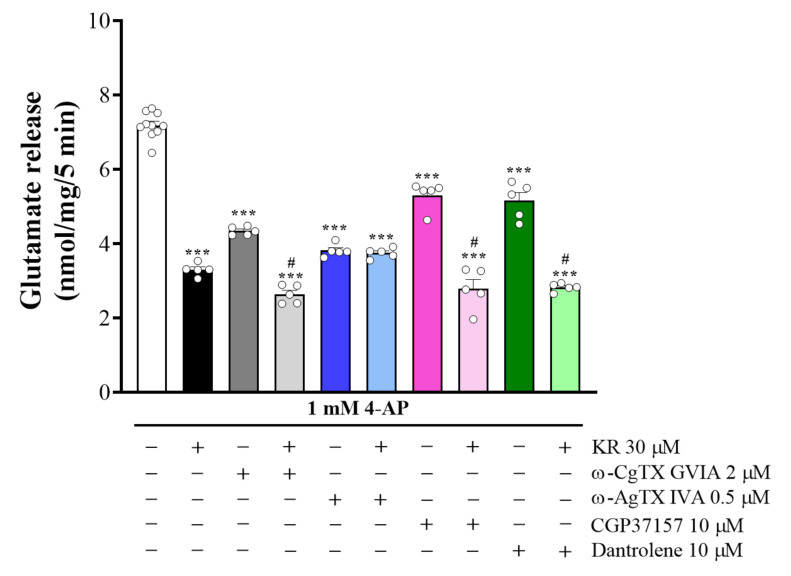
Effects of the N-type Ca^2+^ channel inhibitor ω-CgTXGVIA, P/Q-type Ca^2+^ channel inhibitor ω-AgTXIVA, and intracellular Ca^2+^ release inhibitors dantrolene and CGP37157, on 4-AP-evoked glutamate release in the absence or in the presence of KR. Each bar represents the mean ± S.E.M. of results obtained in 5 experiments. *** *p* < 0.001 versus control; # *p* < 0.001 versus ω-CgTXGVIA, dantrolene or CGP37157 alone.

**Figure 5 molecules-27-01342-f005:**
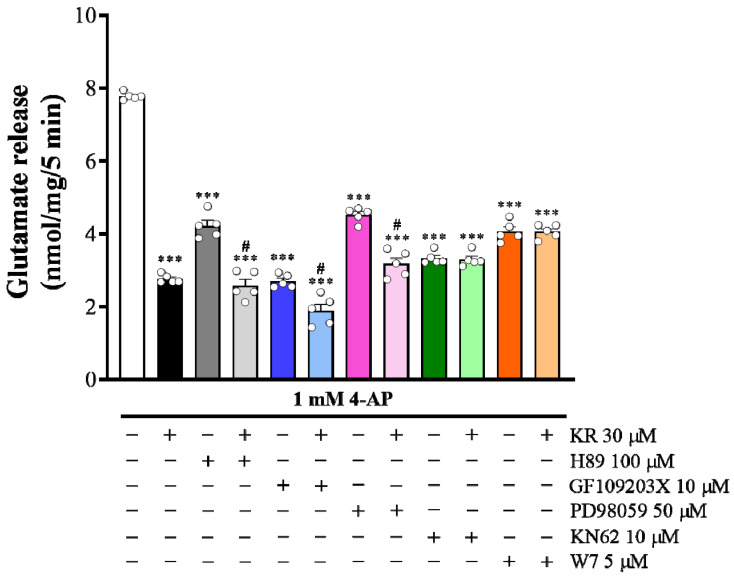
Effects of the PKA inhibitor H89, PKC inhibitor GF109203X, MAPK inhibitor PD98059, CaMKII inhibitor KN62, and calmodulin antagonist W7 on the 4-AP-evoked glutamate release in the absence or in the presence of KR. Each bar represents the mean ± S.E.M. of results obtained in 5 experiments. *** *p* < 0.001 versus control; # *p* < 0.001 versus H89, GF109203X or PD98059 alone.

**Figure 6 molecules-27-01342-f006:**
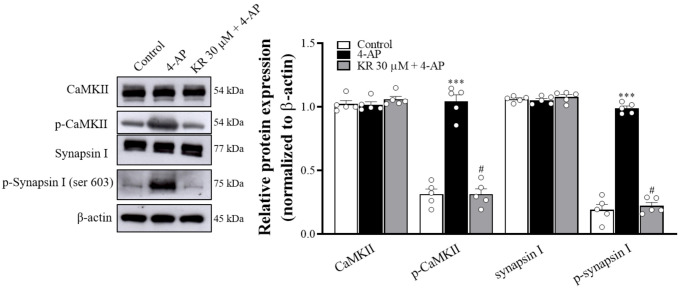
Effect of KR on 4-AP-evoked phosphorylation of CaMKII and synapsin I. KR was added 10 min before the addition of 4-AP (1 mM). Each bar represents the mean ± S.E.M. of results obtained in 5 experiments. *** *p* < 0.001 versus control; # *p* < 0.001 versus 4-AP alone.

**Figure 7 molecules-27-01342-f007:**
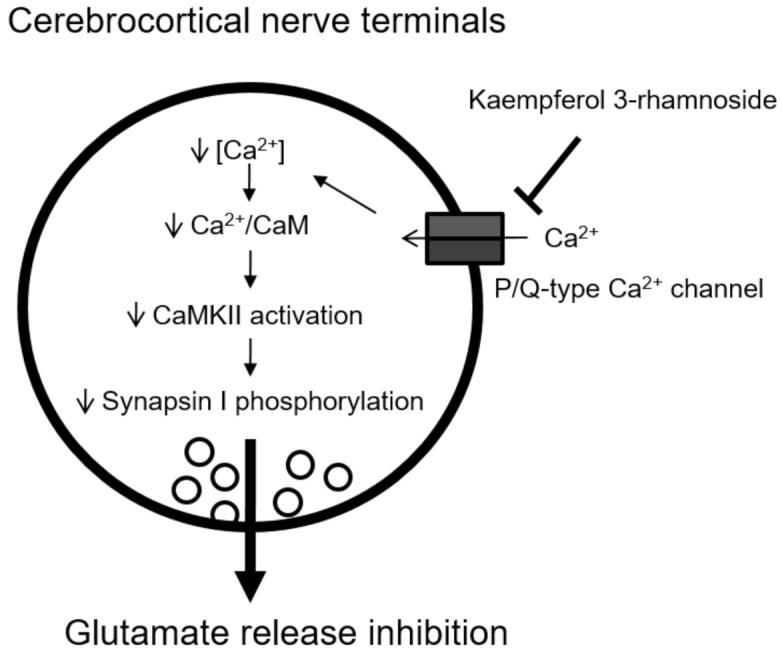
Schematic representation of the main mechanism involved in KR-mediated inhibition of glutamate release from cerebrocortical nerve terminals.

## Data Availability

The data presented in this study are available on request from the corresponding author.

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
