# Peer review of "Kaempferol 3-Rhamnoside on Glutamate Release from Rat Cerebrocortical Nerve Terminals Involves P/Q-Type Ca^2+^ Channel and Ca^2+^/Calmodulin-Dependent Protein Kinase II-Dependent Pathway Suppression"

_molecules, 2022, doi:10.3390/molecules27041342_

Round 1

Reviewer 1 Report

Authors describe results of their work on kaempferol 3-rhamnoside on glutamate release from rat cerebrocortical nerve terminals.

Abstract is missing any basic “digits” supporting conclusions presented therein. Please modify it.

Line 50: put Latin name with italics. (also in vivo/in vitro in the entire manuscript)

Results are comprehensively described and discussed. I have no remarks here. Nice work.

Line 197: check correct name of “Western blotting”

4.1 – give details on how the KR was provided and purity checked.

Conclusions – unlike discussion – are very superficial and don’t show pros and cons of work presented. Neither its importance and applicability in future developments of medical formulae to prevent/slow/fight neurodegenerative diseases.

Author Response

We thank the reviewer for the critical comments and constructive suggestions.

Reviewer 1

Abstract is missing any basic “digits” supporting conclusions presented therein. Please modify it.

Because numerous digits and the limitation of the word number, the digits of the result are not able to be included in the abstract. Hope reviewer can make allowances for this.

Line 50: put Latin name with italics. (also in vivo/in vitro in the entire manuscript)

The word is revised (Page 2, line 50).

Line 197: check correct name of “Western blotting”

The word is corrected (Page 6, line 197).

4.1 – give details on how the KR was provided and purity checked.

About this point, several sentences The bark of Schima superba (2.0 kg) were ground, extracted with methanol at room temperature, and concentrated under reduced pressure to afford a brown resi-due (110 g). This residue was partitioned using EtOAc/H2O, and the EtOAc layer (99 g) was fractionated by silica gel column chromatography using n-hexane-EtOAc, 9:1; n-hexane-EtOAc, 4:1;  n-hexane-EtOAc, 1:1, and n-hexane-EtOAc-MeOH, 1:1:1, to yield four fractions (A–D). Fraction B (102 mg), CH2Cl2-MeOH (4:1) to provide frac-tions B1-B6; fraction B3 (55 mg), CHCl3-MeOH (4:1) to yield KR (10 mg). The identity and purity of KR were verified by proton nuclear magnetic resonance (NMR) spec-troscopy and 2-D NMR spectrometry [36]. are added in the method section (Page 8, lines 270-273; Page 9, lines 274-279). In addition, one reference is included in the present manuscript.

Conclusions – unlike discussion – are very superficial and don’t show pros and cons of work presented. Neither its importance and applicability in future developments of medical formulae to prevent/slow/fight neurodegenerative diseases.

In order to make the statement of the sentence more clear, the sentence in the conclusion section is modified toOur study stimulates its application in preclinical studies directed to the development of a new drug for brain diseases related to glutamate excitotoxicity.(Page 10, Lines 377-379).

Reviewer 2 Report

In my opinion, the article is well written. The authors planned the experiment correctly and has been using a wide range of techniques to characterize the material.

My point on the manuscript is that the origin of kaempferol rhamnoside should be more fully described - provided that the author is not enough. Is it isolated from biological material or synthesized? 

Author Response

We thank the reviewer for the critical comments and constructive suggestions.

Reviewer 2

In my opinion, the article is well written. The authors planned the experiment correctly and has been using a wide range of techniques to characterize the material.

My point on the manuscript is that the origin of kaempferol rhamnoside should be more fully described - provided that the author is not enough. Is it isolated from biological material or synthesized?

About this point, several sentences The bark of Schima superba (2.0 kg) were ground, extracted with methanol at room temperature, and concentrated under reduced pressure to afford a brown resi-due (110 g). This residue was partitioned using EtOAc/H2O, and the EtOAc layer (99 g) was fractionated by silica gel column chromatography using n-hexane-EtOAc, 9:1; n-hexane-EtOAc, 4:1;  n-hexane-EtOAc, 1:1, and n-hexane-EtOAc-MeOH, 1:1:1, to yield four fractions (A–D). Fraction B (102 mg), CH2Cl2-MeOH (4:1) to provide frac-tions B1-B6; fraction B3 (55 mg), CHCl3-MeOH (4:1) to yield KR (10 mg). The identity and purity of KR were verified by proton nuclear magnetic resonance (NMR) spec-troscopy and 2-D NMR spectrometry [36]. are added in the method section (Page 8, lines 270-273; Page 9, lines 274-279). In addition, one reference is included in the present manuscript.

Reviewer 3 Report

The manuscript is well-written and the results support the conclusions.

There are some issues to clarify:

Materials and methods:

The authors should better describe the synaptosome experimental model, with regards to the different phases of perfusion. Yet, they should explain whether the release of the neurotransmitter has been evaluated at a specific time point or it is a total release. Additionally, the authors used the AP K+ blocker, but it is not declared the K+ concentration, in the perfusion medium. Did the authors perfuse synaptosomes with a medium containing K+? If yes, with basal K+ 3 mM medium or with a depolarizing stimulus (12-15 mM)?

Regarding the animal model, how did the authors calculate the animal number for the experiments?

Author Response

We thank the reviewer for the critical comments and constructive suggestions.

Reviewer 3

The manuscript is well-written and the results support the conclusions.

There are some issues to clarify:

Materials and methods:

The authors should better describe the synaptosome experimental model, with regards to the different phases of perfusion. Yet, they should explain whether the release of the neurotransmitter has been evaluated at a specific time point or it is a total release. Additionally, the authors used the AP K+ blocker, but it is not declared the K+ concentration, in the perfusion medium. Did the authors perfuse synaptosomes with a medium containing K+? If yes, with basal K+ 3 mM medium or with a depolarizing stimulus (12-15 mM)?

As suggestion by the reviewer, several sentences are added in the method section, including HEPES buffer medium consisting of 140 mM NaCl, 5 mM KCl, 5 mM NaHCO3, 1 mM MgCl2   6H2O, 1.2 mM Na2HPO4, 10 mM glucose, and 10 mM HEPES at pH 7.4.; Values quoted in the text and expressed in bar graphs represent levels of glutamate cumulatively released after 5 min of depolarization and are expressed as nmol/mg/5 min.(Page 9, lines 307-308, lines 323-325).

Regarding the animal model, how did the authors calculate the animal number for the experiments?

About this point, the sentence is modified to The rats were sacrificed via cervical dislocation to prepare the synaptosomes for en-dogenous glutamate release assay (n=15), cytosolic Ca2+ assay (n=5), membrane po-tential assay (n=5), and Western blotting (n=5).(Page 9, lines 294-297).